# Mitochondrial spongiotic brain disease: astrocytic stress and harmful rapamycin and ketosis effect

Olesia Ignatenko[1], Joni Nikkanen[2], Alexander Kononov[3], Nicola Zamboni[4], Gulayse Ince-Dunn[1], Anu Suomalainen[1,5,6]

Mitochondrial DNA (mtDNA) depletion syndrome (MDS) is a group of severe, tissue-specific diseases of childhood with unknown pathogenesis. Brain-specific MDS manifests as devastating spongiotic encephalopathy with no curative therapy. Here, we report cell type–specific stress responses and effects of rapamycin treatment and ketogenic diet (KD) in mice with spongiotic encephalopathy mimicking human MDS, as these interventions were reported to improve some mitochondrial disease signs or symptoms. These mice with astrocyte-specific knockout of *Twnk* gene encoding replicative mtDNA helicase Twinkle (TwKO[astro]) show wide-spread cell-autonomous astrocyte activation and mitochondrial integrated stress response (ISR[mt]) induction with major metabolic remodeling of the brain. Mice with neuronal-specific TwKO show no ISR[mt]. Both KD and rapamycin lead to rapid deterioration and weight loss of TwKO[astro] and premature trial termination. Although rapamycin had no robust effects on TwKO[astro] brain pathology, KD exacerbated spongiosis, gliosis, and ISR[mt]. Our evidence emphasizes that mitochondrial disease treatments and stress responses are tissue- and disease specific. Furthermore, rapamycin and KD are deleterious in MDS-linked spongiotic encephalopathy, pointing to a crucial role of diet and metabolism for mitochondrial disease progression.

## Introduction

Mitochondrial dysfunction is emerging as a common contributor to different degenerative disorders, including neurodegeneration (Gorman et al, 2016). However, the mechanisms of how mitochondria, the cellular centers of energy metabolism, cause an exceptional variability of non-overlapping diseases remain poorly understood. The tissue specificity is rarely explained by variable expression patterns of mitochondrial proteins, implying that tissues and/or cell types vary in their physiological sensitivity to mitochondrial insults. The disease group lacks curative treatment options.

Mitochondrial DNA (mtDNA) depletion syndrome (MDS) is an excellent example of tissue-specific mitochondrial diseases, manifesting either in the brain, liver, or muscle (Suomalainen & Isohanni, 2010). The causative genes affect nucleotide metabolism or mtDNA maintenance and cause mtDNA loss (Suomalainen & Isohanni, 2010). In the brain, MDS manifests typically as severe epileptic encephalopathy, with cortical laminar necrosis and/or spongiotic encephalopathy (Alpers-Huttenlocher disease; encephalopathic MDS) (Naviaux & Nguyen, 2004). One of the causes of brain-specific MDS are defects of Twinkle, the replicative helicase and copy number regulator of mtDNA (Tyynismaa et al, 2004; Nikali et al, 2005; Hakonen et al, 2008; Ylikallio et al, 2010). Deficiency of functional Twinkle causes mtDNA loss and a consequential loss of mitochondrial RNAs and the core subunits of the mitochondrial respiratory chain complexes I, III, IV and V (Milenkovic et al, 2013; Ignatenko et al, 2018). The molecular mechanisms of Twinkle-linked MDS have remained to be discovered. Previously, we inactivated Twinkle postnatally in mice, in either neurons (TwKO[neuro]) or astrocytes (TwKO[astro]) (Ignatenko et al, 2018). The TwKO[neuro] mice tolerated mtDNA depletion until 7.5–9.5 mo of age without symptoms, and then developed acute, rapidly fatal neurodegeneration. Interestingly, TwKO[astro] mice had a very different disease course. They developed an early-onset neurological disease and progressive spongiotic encephalopathy, with massive cell-autonomous activation of astrocytes (from here on, called cell-autonomous astrogliosis). Such histological findings also characterize brain-specific MDS in children (Alpers, 1931; Sandbank & Lerman, 1972). Classically, reactive gliosis has been attributed to astrocyte response to neuronal pathology (Brown & Squier, 1996; Lake et al, 2015). However, the findings from TwKO[neuro] and TwKO[astro] suggest that astrocytes are the primary affected cell type in spongiotic pathology in MDS. The pathophysiological mechanisms underlying astrocytic reactivity upon mitochondrial dysfunction remain unexplored; however, reactive astrocytes contribute to manifestation of

[1]Stem Cells and Metabolism Research Program, Faculty of Medicine, University of Helsinki, Helsinki, Finland   [2]Cardiovascular Research Institute, University of California, San Francisco, CA, USA   [3]Cancer Research UK, University of Manchester, Manchester, UK   [4]Department of Biology, Institute of Molecular Systems Biology, Eidgenössische Technische Hochschule (ETH) Zurich, Zurich, Switzerland   [5]Neuroscience Center, University of Helsinki, Helsinki, Finland   [6]HUSlab, Helsinki University Hospital, Helsinki, Finland

Correspondence: anu.wartiovaara@helsinki.fi

other neurodegenerative pathologies (Anderson et al, 2016; Liddelow et al, 2017; Yun et al, 2018; Hartmann et al, 2019).

Disease-related stress responses to mitochondrial dysfunction are starting to be elucidated. "Mitochondrial integrated stress response" (ISR$^{mt}$), with a specific transcriptional response led by ATF3-5 transcription factors, and major remodeling of whole-cellular anabolic biosynthesis reactions, is induced by mtDNA expression defects or mitochondrial uncoupling in the skeletal muscle and/or heart (Tyynismaa et al, 2010; Ost et al, 2015; Bao et al, 2016; Nikkanen et al, 2016; Kühl et al, 2017; Restelli et al, 2018; Forsström et al, 2019; Murru et al, 2019). ISR$^{mt}$ includes an early-stage induction of mitochondrial folate cycle (methylenetetrahydrofolate dehydrogenase 2; MTHFD2), fibroblast growth factor 21 (FGF21) and growth/differentiation factor 15 (GDF15), and a second-stage response of serine biosynthesis, transsulfuration pathway, with both autocrine and endocrine metabolic signaling roles of FGF21 (Forsström et al, 2019). The altered metabolite homeostasis leads to imbalanced nucleotide pools and increased levels of serine and glycine (Nikkanen et al, 2016; Forsström et al, 2019; Murru et al, 2019). The mammalian target of rapamycin (mTOR) complex 1 (mTORC1), a nutrient-sensor and a master regulator of cell growth and metabolism, is an upstream controller of ISR$^{mt}$ in the heart and muscle (Nikkanen et al, 2016; Khan et al, 2017). Mouse and human studies suggest specificity of the stress responses to the type of mitochondrial dysfunction or affected cell type (Lehtonen et al, 2016; Khan et al, 2017; Forsström et al, 2019; Murru et al, 2019). However, their relevance to brain diseases remains insufficiently understood.

Rapamycin treatment, inhibiting mTORC1, rescued ISR$^{mt}$ induction and ameliorated pathological hallmarks in the heart and muscle of mitochondrial myopathy mice (Khan et al, 2017). Rapamycin has also been reported to improve disease signs in other mouse models with mitochondrial dysfunctions, including deficiency of respiratory chain complex I (Johnson et al, 2013, 2015; Felici et al, 2017), mitochondrial nucleotide salvage (Siegmund et al, 2017), or muscle complex IV (Civiletto et al, 2018). However, rapamycin enhanced disease progression in mice with coenzyme Q deficiency (Barriocanal-Casado et al, 2019), indicating that not all metabolic defects may benefit from rapamycin. Because of mostly encouraging results, especially in mitochondrial respiratory enzyme defects, rapamycin patient trials have been initiated in children with mitochondrial encephalopathies (https://clinicaltrials.gov/ct2/show/NCT03747328).

Dietary treatments have been proposed to ameliorate mitochondrial disease symptoms. Ketogenic diet (KD) has an anti-epileptic effect and is administered on an elective basis to patients who manifest with drug-resistant epilepsy, including those with mitochondrial disease (Wexler et al, 1997; Kang et al, 2007; Joshi et al, 2009). Cell culture studies suggested that ketone-based nutrition promotes selection against mtDNA deletions (Santra et al, 2004). In mice with mitochondrial myopathy and hepatopathy, KD promoted mitochondrial biogenesis and function (Ahola-Erkkilä et al, 2010; Purhonen et al, 2017). However, in human patients with mitochondrial myopathy and mtDNA deletions, KD induced selective damage of the most affected muscle fibers, with moderate long-term improvement of muscle function (Ahola et al, 2016). These findings indicate that diet composition has a major effect for mitochondrial muscle and liver disease progression.

Here, we report that mtDNA loss in mice activates cell-specific stress responses in mice with either neuronal or astrocytic TwKO.

Furthermore, we explored the efficacy of rapamycin treatment and KD for MDS-related spongiotic encephalopathy. Both interventions led to premature cessation of the treatment trial because of aggravation of disease signs. Our results highlight that mitochondrial dysfunction in different tissues and cell types causes different physiological and treatment responses and emphasize the need of disease-specific treatments based on knowledge of their molecular pathophysiology.

# Results

### Metabolic response to the loss of mtDNA helicase Twinkle in the mouse brain

Twinkle depletion sequentially leads to mtDNA depletion, loss of mtDNA gene expression, and respiratory chain dysfunction (summarized in Fig 1A). MtDNA depletion in neurons causes a late-onset cell death at preterminal stage of 7.5 mo (Ignatenko et al, 2018). mtDNA depletion in astrocytes leads to their cell-autonomous activation, but does not affect the number of astrocytes (Ignatenko et al, 2018). MtDNA loss is more prominent in TwKO$^{neuro}$ brain than in TwKO$^{astro}$ when compared to control mice (Fig S1B), replicating our previous findings. (Ignatenko et al, 2018). TwKO$^{neuro}$, however, appear healthy, whereas TwKO$^{astro}$ mice progressively lose weight starting from 2 to 3 mo of age reaching humane end point (HEP) by the age of 5–6 mo, defined as preterminal stage (Fig S1A), as we also have reported previously (Ignatenko et al, 2018).

To understand the molecular underpinnings of the cell-specific responses to mitochondrial dysfunction, we investigated gene expression and metabolic changes in TwKO$^{astro}$ and TwKO$^{neuro}$ mice (Ignatenko et al, 2018). First, we asked whether ISR$^{mt}$, previously characterized in the skeletal muscle in Twinkle mutant mice (Nikkanen et al, 2016; Khan et al, 2017; Forsström et al, 2019), is also induced in the brain. In 5.5-mo-old TwKO$^{astro}$ mice, we observed an induction of the activating transcription factors (Atf3 and Atf5), serine biosynthesis enzymes (Phgdh and Psat1), mitochondrial folate cycle (Mthfd2), and transsulfuration enzymes (CTH and CBS), as well as the metabolic hormone Gdf15 (Fig 1B–D). Fgf21 expression was not detectable. Neuronal knockout of Twinkle, however, activated none of the tested ISR$^{mt}$-related genes except for Atf3, neither at the preterminal nor young age (7.5 and 3.5 mo, respectively) (Figs 1B–D and S1C).

Next, we evaluated whether ISR$^{mt}$-linked metabolic remodeling occurred in TwKO$^{astro}$ and TwKO$^{neuro}$ mice. Increased serine and glycine levels are typical for ISR$^{mt}$ as a consequence of de novo serine biosynthesis induction (Bao et al, 2016; Nikkanen et al, 2016). Furthermore, altered levels of other proteinogenic amino acids were also linked to mitochondrial dysfunction (Birsoy et al, 2015; Nikkanen et al, 2016; Kühl et al, 2017). Our metabolomics analyses showed a 4.5- and 3.6-fold increase of serine and glycine in TwKO$^{astro}$ (Figs 1E, S1D, and Supplemental Data 1), correlating with an induction of de novo serine biosynthesis enzymes Psat1 and Phgdh (Fig 1B). Interestingly, these findings were not present in TwKO$^{neuro}$ (Figs 1B and E, S1D and E, and Supplemental Data 2). Both models displayed reduced aspartate and increased proline levels (Fig 1E). Several purine precursors or degradation products, also a

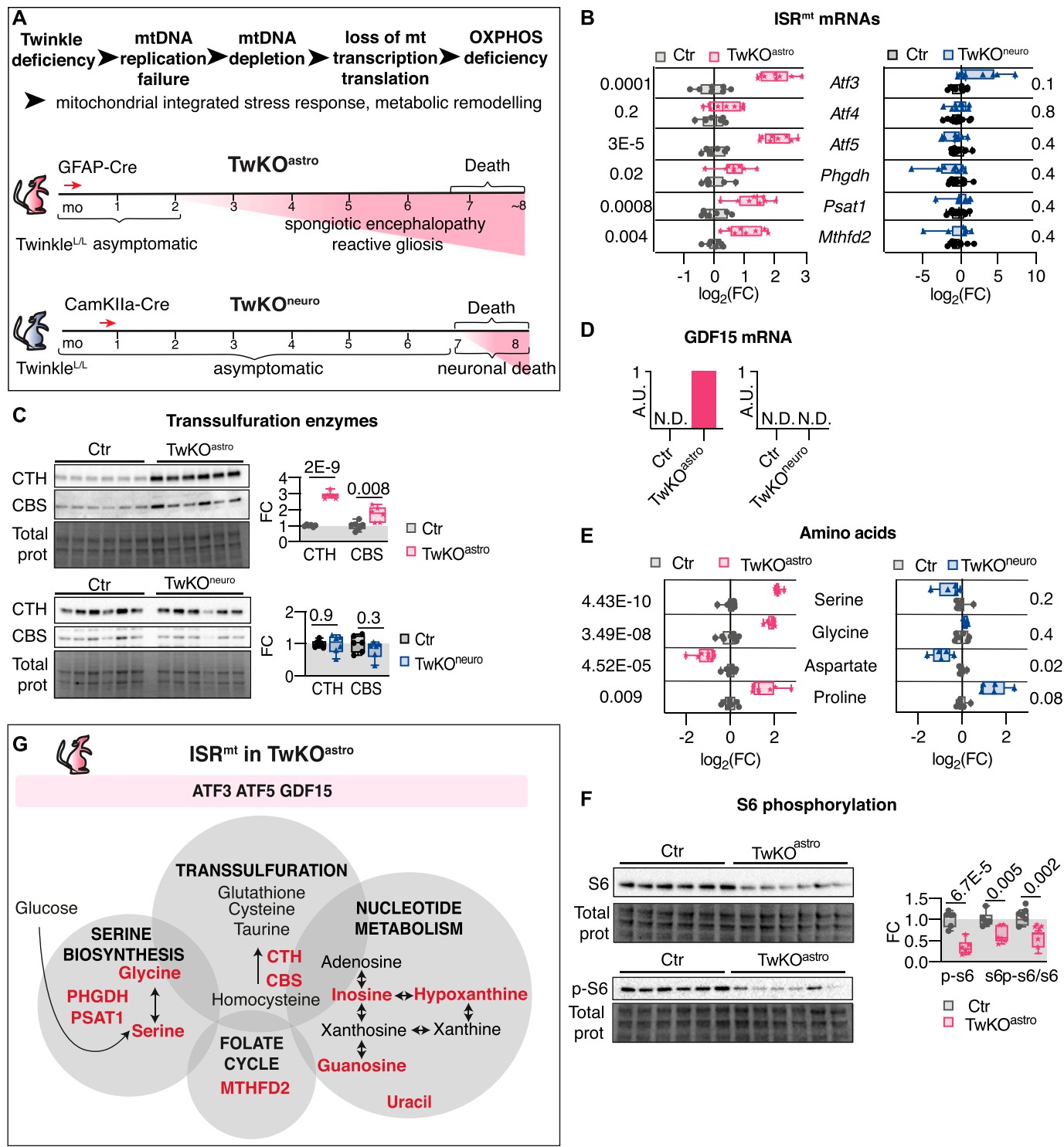

**Figure 1.  ISR^mt is induced in the brain in response to loss of mtDNA helicase Twinkle in astrocytes.**
**(A)** Top: schematic of molecular events upon *Twnk* knockout. Bottom: mouse models used in the study (Ignatenko et al, 2018). **(B)** mRNA expression of ISR^mt genes measured by RT-quantitative PCR. TwKO^astro n = 8 and their controls n = 6–7; TwKO^neuro n = 8–9 and their controls n = 11. **(C)** Immunoblot analysis of transsulfuration pathway enzymes, CTH and CBS. n = 6 per group. **(D)** GFD15 mRNA expression measured by RT-quantitative PCR, A.U.; "1" indicates detectable expression, N.D. indicates no detectable expression. TwKO^astro n = 8 and their controls n = 7, TwKO^neuro n = 5, and their controls n = 5. **(E)** Amino acids, metabolomics. TwKO^astro n = 8 and their controls n = 8, TwKO^neuro n = 5, and their controls n = 6. Multiplicity corrected *P*-values are shown, calculated using the Benjamini–Hochberg procedure. **(F)** Immunoblot analysis of S6 and phospho-S6. n = 6 per group. **(G)** ISR^mt is induced in TwKO^astro brain, schematic representation. Ctr, control mice. For all immunoblots, densitometric quantification of signals is plotted and band intensities are normalized to total protein on the membrane. **(B, C, F)** *P*-values are shown, calculated using unpaired two-tailed parametric *t* test. For all graphs, FC is a fold change ratio, relative to corresponding Ctr mice. For all box and whiskers plots, the box extends from the 25^th to 75^th percentiles; whiskers show all points minimum to maximum; symbols indicate biological replicates. Results of statistical testing can be found in Table S4 and Supplemental Data 1 and 2.

part of ISR$^{mt}$-related metabolic remodeling (Nikkanen et al, 2016), were increased in TwKO$^{astro}$ but not in TwKO$^{neuro}$, and uracil, a pyrimidine metabolite, was increased in TwKO$^{astro}$ (Fig S1F). mTORC1 activation is an integral component of ISR$^{mt}$ in mtDNA maintenance defects in the muscle, showing activated phosphorylation of S6, a ribosomal protein and a target of mTORC1 (Khan et al, 2017). In TwKO$^{astro}$ brains, however, the total S6 amount and p-S6 were remarkably decreased (Fig 1F), suggesting that mTORC1 was not an upstream regulator of ISR$^{mt}$ in TwKO$^{astro}$.

These results indicate that astrocytes activate a robust ISR$^{mt}$ as a response to TwKO, both on a transcriptional and metabolic level, but lacking an induction of mTORC1 or *Fgf21* (Fig 1G). A similar insult in TwKO$^{neuro}$ did not, however, induce ISR$^{mt}$. The evidence highlights cell type specificity of the response in the brain, indicating that metabolic stress in astrocytes activates ISR$^{mt}$.

### Rapamycin does not rescue brain pathology of TwKO$^{astro}$ mice

To directly test the role of mTORC1 in spongiotic encephalopathy progression, we explored the effects of mTORC1 inhibitor rapamycin in TwKO$^{astro}$ mice. We intraperitoneally injected 8 mg/kg/day of rapamycin (Rapa) or vehicle (Veh) into TwKO$^{astro}$ mice, starting at the asymptomatic age of 1.5 mo (Fig 2A). After only 2 wk of treatment, the rapamycin-injected TwKO$^{astro}$ mice started to lose weight reaching a HEP at 2.3 mo of age, when the experiment was terminated (Fig 2B). The baseline mTORC1 activity (S6 phosphorylation) was decreased in the brains of all rapamycin-treated animals, indicating rapamycin access to the brain (Fig 2C). However, genotype-dependent differences were not present (Fig 2C), supporting the previous data that mTORC1 was not part of ISR$^{mt}$ in TwKO$^{astro}$.

At 2.3 mo of age, TwKO$^{astro}$ mice show sparsely located holes in the brain parenchyma, colocalizing with GFAP-positive activated astrocytes (Ignatenko et al, 2018). The progression of spongiosis and reactive gliosis were unaffected by rapamycin (Fig 2D), indicating that mTORC1 activation is not contributing to astrocyte activation or progression of the mitochondrial spongiotic encephalopathy. At the early age of 2.3 mo, TwKO$^{astro}$ did not yet show detectable changes in the components of ISR$^{mt}$, with or without rapamycin treatment (Fig S2A and B). The brain metabolome of TwKO$^{astro}$ did not show a global shift as a consequence of rapamycin treatment, and no evident genotype- or treatment-related clusters were observed (Fig S2C and Supplemental Data 2). However, elevated serine and glycine levels seen in untreated TwKO$^{astro}$ mice were normalized by rapamycin treatment, and the drug also increased methionine levels in TwKO$^{astro}$ mice (Fig S2D). Furthermore, rapamycin increased glutamine levels of both wild-type and TwKO$^{astro}$ mice suggesting its increased production or decreased usage (Fig S2D). Several metabolites showed changes attributable to the genotype (Fig 2E, genotype panel). Rapamycin administration affected the levels of more than 20 metabolites, signifying the overall role of mTORC1 signaling in the brain (Fig 2E, treatment panel). Most of those metabolites were affected similarly in Ctr and TwKO$^{astro}$ mice, implying a similar status of mTORC1 activity in both genotypes. Only three metabolites showed changes attributable to an interaction of the treatment and genotype (Fig 2E, interaction panel). Saccharopine was the most significantly increased metabolite in TwKO$^{astro}$

and was partially rescued by rapamycin treatment (Fig 2F). In mammals, saccharopine is a degradation product of the essential amino acid lysine, and conversion of lysine to saccharopine occurs in the mitochondria (Higashino et al, 1965, 1967). Several lipid metabolites were slightly decreased in TwKO$^{astro}$ brain and were not affected by rapamycin treatment (Fig 2E, genotype panel; Fig 2F).

Finally, we tested whether rapamycin has an effect in TwKO$^{astro}$ brain when started after disease manifestation, at a moderately advanced pathological stage (3 mo, Fig 2G). TwKO$^{astro}$ did not tolerate the treatment well, and their body weight started to decline at already the third day of treatment (Fig 2H). HEP was reached 2 wk after treatment initiation, and the experiment was terminated. However, we found no changes in the brain pathology between the vehicle- and rapamycin-treated mice (Fig 2I). Taken together, rapamycin was not improving the spongiotic pathology and caused rapid deterioration of the mice. The data propose that rapamycin is not a promising treatment strategy for spongiotic encephalopathy because of mtDNA depletion.

### KD worsens the brain pathology of TwKO$^{astro}$

KDs are administered to patients with drug-resistant epilepsy, including patients with mitochondrial diseases, but knowledge of the mechanisms or usefulness in different epileptic states remains limited. We administered KD or regular chow diet for TwKO$^{astro}$ mice after weaning, before disease manifestation (Fig 3A). KD increased blood β-hydroxybutyrate levels, indicating efficient ketosis (Fig 3B). Weight progression slowed down both in Ctr and TwKO$^{astro}$ mice (Fig 3C), and HEP of TwKO$^{astro}$ body weight loss was reached after 9.5 wk of treatment, requiring termination of the mice (Fig 3C). KD remarkably aggravated brain pathology of TwKO$^{astro}$ mice. The spongiosis and gliosis were more severe than those in mice fed with regular chow diet (Fig 3D). Of ISR$^{mt}$ transcripts, *Atf3* and *Atf5* showed further induction (Fig S3A), and transsulfuration enzymes levels were increased in KD-fed TwKO$^{astro}$ mice (Fig 3E). *Gdf15* mRNA was detectable in all TwKO$^{astro}$ mice but not in the Ctr mice, with high variation in KD-fed TwKO$^{astro}$ mice (Fig S3B).

Analysis of brain metabolome revealed major metabolic reprogramming of TwKO$^{astro}$ brain at 3.2 mo of age (Fig S3C and Supplemental Data 2). 128 out of 382 detected metabolites showed changes attributable to the genotype (Figs 3F and S3D), whereas only seven were changed at 2.3 mo of age (Fig 2E). KD modified the metabolome of both TwKO$^{astro}$ and Ctr brain (Fig S3D, treatment panel). In TwKO$^{astro}$, the level of saccharopine, the most significantly changed metabolite at the early disease stage in TwKO$^{astro}$ (Fig 2F), was further increased in KD-fed TwKO$^{astro}$ (Fig 3G). KD also boosted levels of serine and glycine in TwKO$^{astro}$, indicating further up-regulation of ISR$^{mt}$ in treated mice, despite the fact that de novo serine synthesis enzyme expression levels were not detectably induced at this age (Figs 3G and S3A). KD increased methionine levels in Ctr but not in TwKO$^{astro}$ mice (Fig S3E). Levels of other proteogenic amino acids were mostly unchanged in TwKO$^{astro}$ mice (Fig S3E). Aspartate, produced in the mitochondria, was decreased by KD in TwKO$^{astro}$, potentially reducing overall translation (Fig 3G). ATP, ADP, UDP, and FAD declined after KD in TwKO$^{astro}$ but not in Ctr

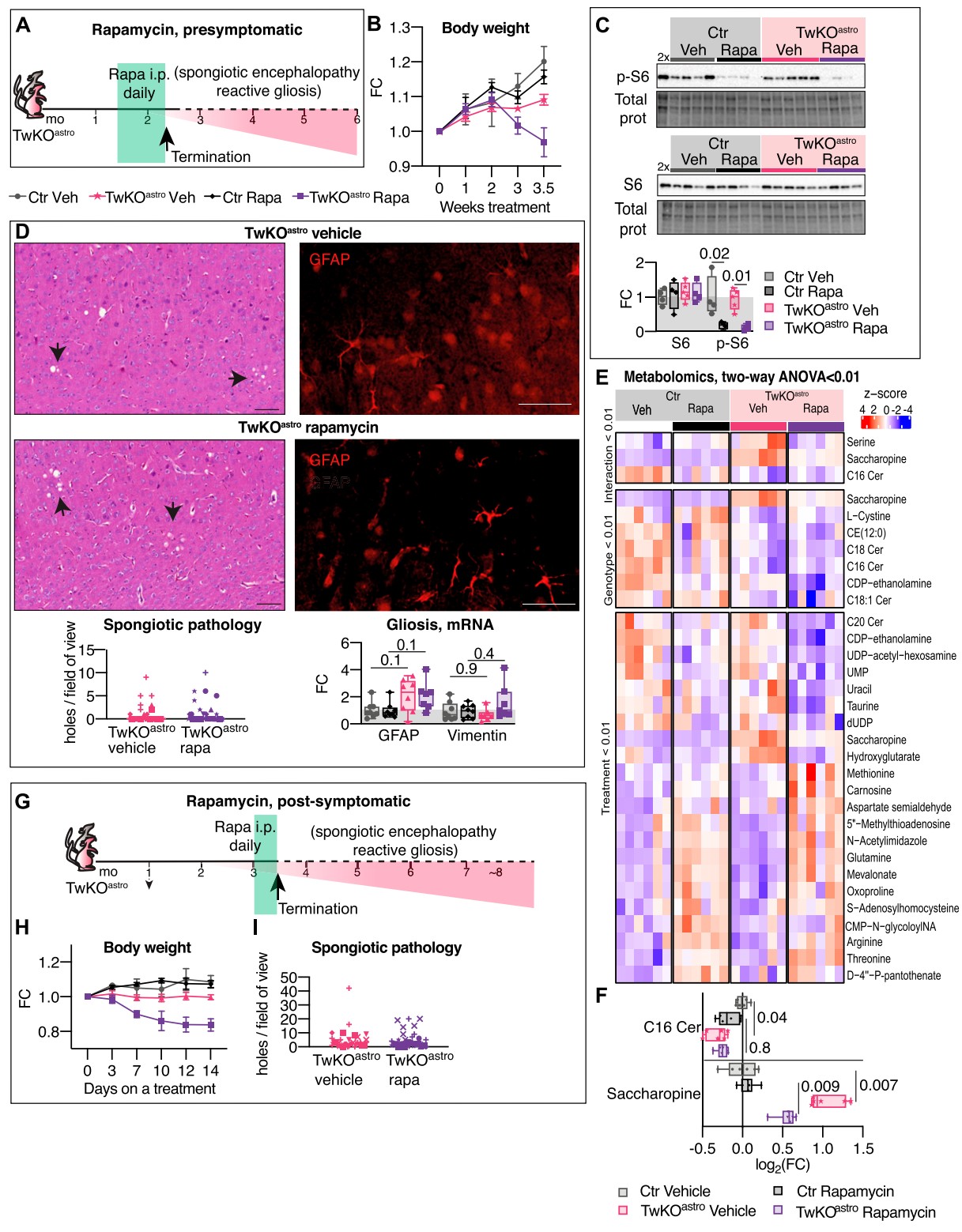

**Figure 2. Rapamycin does not prevent or rescue brain pathology of TwKO^astro.**

**(A, B, C, D, E, F, G, H, I)** Rapamycin treatment started at presymptomatic (A, B, C, D, E, F) and at a moderately advanced (G, H, I) phenotype stage. **(A)** Schematic of presymptomatic rapamycin treatment. Daily i.p. injections of rapamycin or vehicle were initiated at 1.5 mo and were continued until humane end point in body weight loss was reached (2.3 mo, arrow). Green area shows treatment period and pink area represents symptom progression in untreated TwKO^astro mice. **(B)** Body weight progression upon rapamycin treatment initiated at the presymptomatic stage. Weight change is shown as a fold change (FC) relative to the body weight at treatment initiation (zero time point). n = 9 per group. Mean with SD is shown. **(C)** Immunoblot analysis of total S6 and phospho-S6 proteins. Ctr Veh n = 4, Ctr Rapa n = 4, TwKO^astro Rapa n = 4,

(Fig 3F), indicating an imbalance of nucleotide metabolism and high-energy phosphate donors in diseased mice.

Altogether, our evidence indicates a remarkable effect of diet and nutrient composition to the progression of mitochondrial spongiotic encephalopathy, with deleterious consequences of both rapamycin and KD.

## Discussion

How different diseases and treatments affect metabolism in distinct tissues and cell types is a central question concerning mechanisms of tissue specificity and treatment effects. The topic is especially important for mitochondrial diseases that show an exceptional variability of manifestations, despite the primary origin being in one organelle. Here, we demonstrate that astrocytes and neurons respond differently to metabolic stress caused by TwKO and mtDNA loss. Astrocyte-specific TwKO leads to a robust mitochondrial integrated stress response, associated with cell-autonomous activation of the astrocytes, whereas in neurons, a similar stress causes few signs of ISR$^{mt}$. Furthermore, we show that metabolic modifier interventions, rapamycin and KD, are deleterious for mice manifesting with mitochondrial spongiotic encephalopathy related to mtDNA depletion syndrome. Our results highlight the context-dependence of stress and treatment responses, even when different cell types encounter similar stresses.

Mitochondrial transcriptome and proteome in astrocytes and neurons differ (Eraso-Pichot et al, 2018; Fecher et al, 2019), indicating specialization of mitochondrial functions to distinct cell types of the brain. We show here that mtDNA depletion in astrocytes causes ISR$^{mt}$ that closely resembles the response reported in the skeletal muscle and heart with mtDNA expression insult (Nikkanen et al, 2016; Khan et al, 2017; Kühl et al, 2017; Forsström et al, 2019) or mitochondrial uncoupling (Ost et al, 2015). Astrocytic but not neuronal mtDNA depletion induced de novo serine biosynthesis and transsulfuration pathway enzyme expression. Previous reports found that in a healthy brain, the expression of these enzymes is enriched in astrocytes compared with other brain cell types, which possibly also explains the differential regulation of these pathways upon mtDNA loss (Zhang et al, 2014; Chai et al, 2017; Diaz-Castro et al,

2019). The signals for the induction are unknown, but could involve amino acids, metabolites, ATP, redox-status, gasotransmitter hydrogen sulfide, and/or glutathione synthesis. In the Twinkle-linked mtDNA expression defect manifesting as mitochondrial myopathy, and in cultured cancer cells, a similar ISR$^{mt}$ response drove glucose carbon flux to glutathione synthesis (Locasale, 2013; Nikkanen et al, 2016) and to NADH production (Yang et al, 2020). Which of the pleiotropic glucose carbon fates are crucial for astrocytes with MDS remains unknown.

Specific changes in amino acid metabolism were found in TwKO$^{astro}$ mice. Serine and glycine amounts were increased, consistent with induced serine biosynthesis. Proline amount was increased both in TwKO$^{neuro}$ and TwKO$^{astro}$ mice. Proline is oxidized by proline dehydrogenase, which is enriched in astrocyte mitochondria (Zhang et al, 2014), and contributes to NADP/NADPH recycling between mitochondria and the cytosol, as well as ATP production upon nutrient stress. Proline metabolism was suggested to be part of ISR$^{mt}$ in different mtDNA expression defects in the heart (Kühl et al, 2017), thereby indicating this pathway to be widely remodeled upon mitochondrial dysfunction in different cell types. Aspartate level was reduced in TwKO$^{astro}$ and was also previously reported to be a marker of mitochondrial dysfunction in cancer cells (Birsoy et al, 2015). Overall, our evidence highlights the central role of astrocyte mitochondria in amino acid metabolism in the brain.

Saccharopine is an intermediate of lysine metabolism in mammalian tissues (Hallen et al, 2013), including cultured human astrocytes and neuronal progenitors (Crowther et al, 2019) and was one of the most significantly up-regulated metabolites in TwKO$^{astro}$ brain. Lysine degradation functions as an antioxidative pathway in yeast (Olin-Sandoval et al, 2019); however, saccharopine accumulation causes swelling of mitochondria in *Caenorhabditis elegans* and mice (Zhou et al, 2019). Our data point to saccharopine to be a novel, early marker of mitochondrial dysfunction in the brain, the contributions of which to pathogenesis remain to be studied.

mTORC1 is a major regulator of amino acid metabolism and has also been reported to control the key components of ISR$^{mt}$ (transsulfuration, mitochondrial folate cycle, nucleotide synthesis, and serine biosynthesis) in human cancer cells (Ben-Sahra et al, 2016) and in the heart and muscle in mitochondrial myopathy mice (Khan et al, 2017). However, we found no signs of mTORC1 activation in the brain of TwKO$^{astro}$, which otherwise robustly up-regulated

TwKO$^{astro}$ Veh n = 5. 2× is a Ctr Veh sample, with a twofold amount of protein loaded on the lane compared with other samples. **(D)** Brain pathology, rapamycin treatment. Left: Representative cortical sections show spongiotic encephalopathy. Hematoxylin and eosin staining of TwKO$^{astro}$ treated with vehicle or rapamycin. Arrows show sponge-like holes typical for spongiotic encephalopathy. Quantification below: counts of holes per field of view, n = 7 mice per group, two sections per mouse, three fields of view per section. Counts from the fields of view of one mouse are plotted with the same symbol. Scale bars, 50 μm. Right: Astrocyte activation. Glial fibrillary acidic protein (GFAP), immunofluorescent detection, cortex, representative of n = 7 mice per group, two sections per mouse. Below: mRNA expression of GFAP and vimentin (reactive gliosis markers) measured by RT-quantitative PCR. Ctr Veh n = 8, Ctr Rapa n = 8, TwKO$^{astro}$ Veh n = 7–8, TwKO$^{astro}$ Rapa n = 7. **(E)** Metabolomics analysis. All metabolites with *P*-value < 0.01 in any of the two-way ANOVA tests (interaction, genotype, and treatment) are plotted. Z-scores are calculated from intensity values. Note that metabolite is plotted more than once if it passes selection criteria in more than one test. n = 6 per group. Full data set is shown in Fig S2C. **(F)** Metabolomics, full data set is shown in Fig S2C. n = 6 per group. Multiplicity corrected *P*-values are shown, calculated using the Benjamini–Hochberg procedure. **(G, H, I)** Rapamycin treatment started at a moderately advanced phenotype stage. **(G)** Schematic of the rapamycin treatment at a moderately advanced disease stage of TwKO$^{astro}$ mice. Daily i.p. injections of rapamycin or vehicle were started at a moderately advanced phenotype stage (3 mo) and were continued until humane end point in body weight loss was reached (3.5 mo, arrow). Green area shows treatment period and the pink area represents symptom progression in untreated TwKO$^{astro}$ mice. **(H)** Body weight progression upon rapamycin treatment initiated at a moderately advanced disease stage. Ctr Veh n = 3, Ctr Rapa n = 5, TwKO$^{astro}$ Veh n = 7, TwKO$^{astro}$ Rapa n = 8. Weight change is shown as an FC relative to the body weight at treatment initiation (zero time point). Mean with SD is shown. **(I)** Quantification of spongiotic encephalopathy. Counts of holes per field of view in brain sections stained with hematoxylin and eosin, 1–2 sections per mouse, three fields of view per section. Counts from fields of view of the same mouse are plotted with the same symbol. TwKO$^{astro}$ Veh n = 6, TwKO$^{astro}$ Rapa n = 8. Ctr, control mice; Veh, vehicle-treated mice; Rapa, rapamycin-treated mice. **(C, D)** *P*-values are shown, calculated using two-way ANOVA followed by Tukey correction for multiple comparisons. For all graphs, FC is a fold change ratio, relative to corresponding Ctr mice. For all box and whiskers plots, the box extends from the 25th to 75th percentiles; whiskers show all points minimum to maximum; symbols indicate biological replicates. Results of statistical testing can be found in Table S4 and Supplemental Data 2.

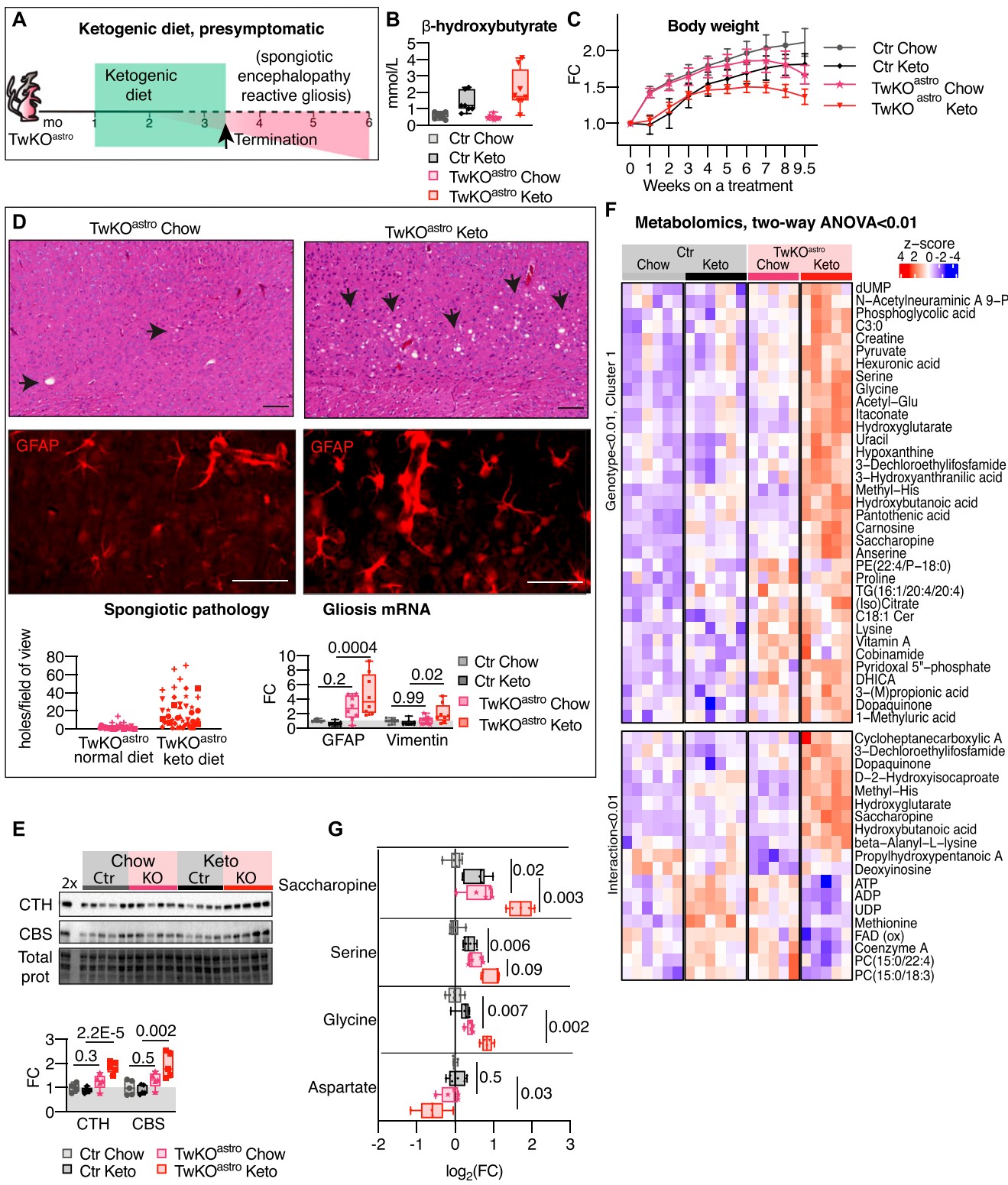

**Figure 3. Ketogenic diet accelerates brain pathology of TwKO[astro].**

Ketogenic diet started upon weaning of the mice. **(A)** Schematic representation of the ketogenic diet experiment. Ketogenic or chow diet was administered to the mice upon weaning. Experiment was continued until humane end point body weight loss of TwKO[astro] was reached (3.2 mo, arrow). Green area shows a treatment period and pink area represents a symptom progression in untreated TwKO[astro] mice. **(B)** Blood β-hydroxybutyrate levels 8 wk after start of the experiment. Ctr Chow n = 10, Ctr Keto n = 7, TwKO[astro] Chow n = 9, TwKO[astro] Keto n = 8. **(C)** Body weight progression. Ctr Chow n = 12, Ctr Keto n = 8, TwKO[astro] Chow n = 16, TwKO[astro] Keto n = 14. Weight change is shown as fold change (FC) relative to the zero time point. Mean with SD is shown. **(D)** Brain pathology, ketogenic diet. Left: Spongiotic encephalopathy. Representative

ISR[mt] components. Rapamycin treatment did not affect brain pathology of TwKO[astro] but caused accelerated body weight loss leading to a premature cessation of the experiment. The harmful effect was not expected, as previous reports suggested beneficial effects of rapamycin on different mouse models with mitochondrial dysfunction: in NDUFS4-KO (Johnson et al, 2013, 2015; Felici et al, 2017), thymidine kinase-2 knock-in (Siegmund et al, 2017), and muscle-specific COX15-KO (complex IV assembly deficiency) (Civiletto et al, 2018) mice. Recently, rapamycin failed to rescue the brain pathology of mice with coenzyme Q deficiency, characterized by spongiotic encephalopathy and astrogliosis, similar to TwKO[astro] mice (García-Corzo et al, 2013; Barriocanal-Casado et al, 2019). In chow-fed TwKO[astro], the levels of ubiquinone-1 and ubiquinone-2 are decreased at the age of 3.2 mo, suggesting a secondary coenzyme Q deficiency (Supplemental Data 2). Whether similar metabolic changes in astrocytes underlie spongiosis and negative rapamycin effect in CoQ deficiency and TwKO[astro] remains to be studied. These results indicate that the effect of rapamycin depends on the underlying primary defect and may be deleterious in mitochondrial spongiotic encephalopathy. Rapamycin has profound effects on organismal metabolism in general, which may also contribute to the worsening of the phenotype, even in an encephalopathic model.

KD was also poorly tolerated by TwKO[astro], and different from rapamycin, it accelerated the brain pathology progression. Astrocytes ensure nutrient uptake from the blood flow and are believed to survive rather severe mitochondrial dysfunction because of their capacity to up-regulate glycolysis (Supplie et al, 2017). KD, however, minimizes glucose availability for glycolytic ATP production. Despite this knowledge, KD has been used to treat epilepsy in mitochondrial diseases (Wexler et al, 1997; Kang et al, 2007; Joshi et al, 2009). Our findings present a proof-of-principle that KD worsens spongiotic encephalopathy caused by mitochondrial dysfunction in astrocytes.

In conclusion, the exceptional heterogeneity and tissue specificity of different mitochondrial diseases indicate that mitochondrial functions are cell type specific. Therefore, also the pathophysiological changes may drastically vary in different diseases, and require specific treatments. Promising results of interventions, such as rapamycin and KD in some mitochondrial diseases, are not automatically generalizable to the whole disease group. Our preclinical data indicate deleterious effects of rapamycin and KD in brain-related MDS, with rapid deterioration of general condition and/or brain pathology of the mice. Therefore, our evidence does not support the use of these intervention strategies for mitochondrial spongiotic encephalopathy or epilepsy linked with mtDNA depletion.

Our results indicate that metabolic modifier therapies and diets are potent modifiers of disease progression.

### Limitations of the study

Usage of genetically modified mouse models with cell type–specific gene knockouts provided important insights into the physiology of tissues in normal and pathological conditions. However, comparing different Cre-induced knockouts with each other is limited by the absence of strict temporal control of the effect of knockout, that is, different rates of loss of the targeted product in different cells or tissues, as well as differently tolerated thresholds of such loss. Most, if not all, genes showing exclusive or highly enriched expression in astrocytes in brain tissue are also expressed at some level by peripheral organs. GFAP73.12-Cre used to generate TwKO[astro] is expressed by subpopulations of neurons in the mouse cortex, and by specific peripheral tissues (Bush et al, 1998; Smith et al, 2019). Such expression was unlikely to affect the findings, as TwKO[neuro] showed no responses even in the terminal stage, as these were found in the TwKO[astro] soon after symptom manifestation, and the spongiotic encephalopathy observed in TwKO[astro] is a very distinct finding. Future studies are needed to study the relevance of our findings for human MDS manifesting with morphologically similar spongiotic pathology.

# Materials and Methods

### Animal studies

Animal experiments were approved by The National Animal Experiment Review Board and Regional State Administrative Agency for Southern Finland, following the European Union Directive. Mice were maintained in a vivarium with 12-h light:dark cycle at 22°C. Mice were fed with Altromin 1324 diet and water ad libitum, unless otherwise specified. C57Bl/6OlaHsd or mixed genetic background mice were used (mixed background originates from mT/mG reporter mice). TwKO[astro] and TwKO[neuro] models are derived from breeding mice carrying loxP sites in *Twnk* gene (Tw[loxp/loxp]) to (Gfap73.12-Cre Tw[loxp/loxp]) or (CamKII-Cre Tw[loxp/loxp]), respectively. Gfap73.12-Cre, CamKII-Cre, and mT/mG were originally received from The Jackson Laboratory and maintained in the local mouse facility (Stock No: 005359, 012886, 00757). Tw[loxp/loxp] mice carry Y508C mutation in the targeted *Twnk* gene (Nikkanen et al, 2016).

---

cortical sections, hematoxylin and eosin staining of TwKO[astro] fed with chow or ketogenic diet. Arrows show sponge-like holes typical for spongiotic encephalopathy. Quantification below: counts of holes per field of view, TwKO[astro] Chow n = 6, TwKO[astro] Keto = 8, one to two sections per mouse, three fields of view per section. Counts from the fields of view of one mouse are plotted with the same symbol. Scale bars, 50 $\mu$m. Right: Astrogliosis. GFAP, immunofluorescent detection, cortex, representative of n = 8 mice per group, two sections per mouse. Below: mRNA expression of GFAP and vimentin (astrocyte activation markers) measured using RT-quantitative PCR. Ctr Chow n = 7, Ctr Keto n = 8, TwKO[astro] Chow n = 8, TwKO[astro] Keto n = 8. **(E)** Immunoblot analysis of transsulfuration pathway enzymes CTH and CBS. Densitometry of Western blot signals: a protein signal normalized to the total protein in the membrane. n = 5 per group. 2× is a Ctr Chow sample with a twofold amount of protein loaded on the lane, compared with other samples. **(F)** Metabolomics analysis. Metabolites with $P$-value < 0.01 in the two-way ANOVA interaction tests, and a cluster of metabolites with $P$-value < 0.01 in the two-way ANOVA genotypes tests are plotted (see selection of the cluster in Fig S3D). Z-scores are calculated from intensity values. Note that metabolite is plotted more than once if it passes selection criteria in more than one test. Ctr Chow, Ctr Keto n = 6; TwKO[astro] Chow, TwKO[astro] Keto n = 5. Full data set is shown in Fig S3C. **(G)** Metabolomics, full data set is shown in Fig S3C. Ctr Chow n = 6, Ctr Keto n = 6, TwKO[astro] Chow n = 5, TwKO[astro] Keto n = 5. Multiplicity corrected $P$-values are shown, calculated using the Benjamini–Hochberg procedure. Ctr, control mice; Chow, mice fed with chow diet; Keto, mice fed with ketogenic diet. For all graphs, FC is a fold change ratio, relative to corresponding Ctr mice. For all box and whiskers plots, the box extends from the 25th to 75th percentiles; whiskers show all points minimum to maximum; symbols indicate biological replicates. **(C, D, E)** $P$-values are shown, calculated using two-way ANOVA followed by Tukey correction for multiple comparisons. Results of statistical testing can be found in Table S4 and Supplemental Data 2.

Tw$^{loxp/loxp}$ and Tw$^{loxp/null}$ littermates were used as controls, which are asymptomatic and indistinguishable from wild-type mice. The preterminal stage is defined upon reaching HEP as follows: 7.5-mo-old TwKO$^{neuro}$ (before or upon severe manifestation preceding death of otherwise asymptomatic mice) and 5.5-mo-old TwKO$^{astro}$ (due to body weight loss, otherwise lethal at 7–8 mo as described previously) (Ignatenko et al, 2018). Exact ages and sexes of mice used in the study can be found in Table S1.

## Rapamycin treatment

Rapamycin (#R-5000; LC Laboratories) was dissolved in DMSO to 100 mg/ml, then further diluted in 5% PEG-400/5% Tween 20 to a final concentration of 1.6 mg/ml, sterile filtered, and stored at –80°C. Mice were weighed individually, and 8 mg/kg of body weight of rapamycin was injected daily intraperitoneally at the indicated dosage, previously shown to be effective in the brain (Johnson et al, 2013), as in Khan et al (2017). Control mice were injected with the same volume of diluent without rapamycin.

## KD

Mice were administered upon weaning with Teklad Custom Diet TD.170470 containing short- and medium-chain fatty acids; with coconut oil, milk fat, and soybean oil as fat sources. 90.5% kcal is derived from fat, 9.2% from protein, and 0.3% from carbohydrates. The ratio of fat to protein + carbohydrate is ~4.25. KD was administered on a petri dish to the bottom of the cage and changed every 3–4 d. Control mice were administered with an Altromin 1324 chow diet. Mice which failed to accept the diet and lost body weight upon initiating the diet were terminated and not included in the study. β-hydroxybutyrate levels were measured as follows: food was removed from the cage for 4 h, and β-hydroxybutyrate was measured from a tail vein drop of blood with Freestyle Precision b-Ketone, Abbott.

## Tissue collection

Mice were euthanized with $CO_2$, blood was collected by cardiac puncture, and mice were decapitated. Brain was removed from the skull and placed into an acrylic brain slicer. Sagittal cut along the midline was made, and one 3-mm hemisphere slice was collected into 4% PFA in PBS fixative for histological experiments, post-fixed, dehydrated, and paraffin-embedded. The second hemisphere was dissected, and pieces of cortex were snap-frozen in liquid nitrogen.

## RT quantitative PCR

15–30 mg of snap-frozen mouse brain cortex was lysed in QIAzol lysis reagent (QIAGEN) in soft tissue homogenizing tubes with ceramic beads (Precellys) using Precellys 24 homogenizer (Precellys). Total RNA was extracted using the RNeasy kit (QIAGEN). RNA concentration was measured using the NanoDrop1000 Spectrophotometer. Maxima first-strand cDNA synthesis kit with dsDNAse (Thermo Fisher Scientific) was used to eliminate genomic DNA and to generate cDNA from 600 to 1,500 ng of RNA. cDNA was diluted to 1/20-1/50, and 1 μl of diluted cDNA was used for RT-quantitative

PCR reactions performed with IQ SybrGreen kit (Bio-Rad) or SensiFast SYBR kit (Bioline). Oligonucleotides were designed to amplify 70–150-bp-long product and to be separated by at least one intron of the corresponding genomic DNA of a minimum 1,000 bp length (whenever possible) using NCBI primer BLAST software (Ye et al, 2012). Oligonucleotides with 85–105% efficiency which amplifies one product of expected length were selected. All oligonucleotide sequences can be found in Table S2. The expression of the genes of interest was normalized to hydroxymethylbilane synthase (HMBS) expression.

## MtDNA copy number analysis

15–30 mg of snap-frozen mouse brain was lysed in TNES buffer with Proteinase K at 55°C in a water bath overnight. DNA was extracted with phenol–chloroform extraction followed by ethanol and ammonium acetate precipitation. Pellet was washed with ethanol, dried, and resuspended in Tris–HCl. Analysis of mtDNA was performed by quantitative PCR and normalized to a nuclear gene; primer sequences can be found in Table S2.

## Western blotting

15–30 mg of snap-frozen mouse brain was solubilized in a buffer containing 50 mM Tris–HCL, pH 7.5, 150 mM NaCl, 1% Triton X-100, protease inhibitor cocktail (Roche), and phosphatase inhibitor cocktail (Thermo Fisher Scientific). Protein concentrations were measured by the Bradford Assay (Bio-Rad), and the concentration was adjusted to 1 μg/ul in Laemmli sample buffer. Equal amounts (typically, 7.5 μg) of proteins were loaded on Criterion or mini-PROTEAN TGX Stain-Free 4–20% precast gels (Bio-Rad), separated by Tris-glycine SDS–PAGE and transferred to PVDF membranes using Trans-Blot Turbo Transfer System (Bio-Rad). BlueStar Plus pre-stained protein marker was used to estimate the molecular weight of the proteins. Stain-free membrane images upon transfer completion were used for the normalization of protein chemiluminescent signal. Membranes were blocked in Tris-buffered saline with 0.1% Tween (TBST) with 4% milk for 1 h at room temperature. Primary antibodies were incubated overnight at +4°C, diluted in 1.5% BSA/TBST or 4% milk/TBST, and detected the following day with secondary HRP-conjugated antibodies using ECL (Bio-Rad). Unsaturated images were collected with a ChemiDoc detection system. Images for the main figures were cropped in Adobe In Design or Adobe Photoshop, and linear adjustments of brightness and contrast were applied equally across the entire image. Original photographs of Western blot membranes can be found in Fig S4. List of used antibodies can be found in Table S3. Densitometry analysis was performed in ImageLab software (Bio-Rad).

## Histology

Histological and immunofluorescent assays were carried on 7–10-μm deparaffinized sagittal sections.

### Hematoxylin and eosin
Brain sections were stained with hematoxylin and eosin. 1–2 brain sections per mouse were imaged and digitalized with Pannoramic

Digital Slide Scanner (3DHISTECH). Three snapshots of cortical area per section were acquired in CaseViewer software by an investigator blind to the genotypes and treatments of the mice.

### Spongiotic pathology
Spongiotic pathology was quantified from hematoxylin and eosin stain images acquired by an investigator blind to genotypes and treatments. Counting was performed in Fiji software by marking round-shaped holes corresponding to spongiotic pathology by an investigator blind to the treatments of the mice.

### Immunofluorescence
Antigen retrieval by heating in 10 mM citric acid was performed on brain sections. Samples were blocked by incubation in the following blocking solution: 10% horse serum and 0.1% Triton X-100 in PBS at room temperature. Consequently, sections were incubated overnight at +4°C with primary antibodies diluted in blocking solution. Next day, the sections were incubated with secondary antibodies conjugated with Alexa Fluor fluorescent probes (Thermo Fisher Scientific) diluted in blocking solution and mounted with DAPI-containing mounting medium (Vectashield). Images were acquired with a Zeiss Axio Imager epifluorescent microscope, and only linear adjustments were applied. A list of used antibodies can be found in Table S3.

### Metabolomics

### Targeted metabolomics
Targeted metabolomics was performed using Waters Acquity ultra-performance liquid chromatography and triple-quadrupole mass spectrometry analysis in FIMM (Institute for Molecular Medicine Finland) as in Nikkanen et al (2016), Khan et al (2017) and Forsström et al (2019). The polar metabolome was extracted from frozen 20 mg mouse muscle and lysed with ceramic beads (Precellys) containing labeled internal standard mix and extraction solvent. The exact procedure and its validation are described in Nandania et al (2018). Quantification was performed by internal standards and external calibration curves.

### Untargeted metabolomics
20–30 mg of snap-frozen mouse brain cortex was homogenized in 70% ethanol cold extraction solvent (handled in ethanol cold bath with dry ice, approximately –40°C) with ceramic beads using a Precellys 24 homogenizer (Precellys). The polar metabolome was extracted with 70% ethanol in water (prewarmed) at 75°C for 1 min. The clear extracts were analyzed by flow injection analysis on an Agilent 6550 QTOF instrument (Fuhrer et al, 2011) in negative mode. Putative annotations were made searching against the Human Metabolome Database (version 3.6) using accurate mass (tolerance 0.001 m/z) and isotopic patterns. While this approach is sufficient to assign molecular formulas in most cases, it does not allow distinguishing between isomers. For simplicity, only one name is reported in the heat map, but the full annotation is provided in Supplemental Data 2.

### Statistical analysis

Statistical analysis of data presented in the figures is reported in the Table S4, and was performed using Prism 8 and Excel (Microsoft); graphs were made with Prism 8 software. A comparison of two groups (Figs 1 and S1) was performed using unpaired two-tailed parametric *t* test. Analysis of treatment experiments (Figs 2, 3, S2, and S3) was performed using two-way ANOVA and Tukey correction for multiple comparisons (Prism 8). Statistical analysis of metabolomics data can be found in Supplemental Data 1 and 2 and was performed in R using two-tailed parametric *t* test to calculate the *P*-value followed by false discovery rate–based methods of *P*-value calculation: *p.adjusted* value was calculated using the Benjamini–Hochberg procedure (Benjamini & Hochberg, 1995); *qvalue* was calculated using Storey and Tibshirani (2003) and Storey et al (2020). Heat maps were generated using R (Gu et al, 2016), and clustering was supervised by genotypes and treatments. To find metabolites identified in both targeted and untargeted metabolomics data sets, HMDB identifiers were used (Figs 1E and S1E).

## Data Availability

Full metabolomics data are available as supplementary data sets.

## Supplementary Information

## Acknowledgements

We thank Kirsi Mattinen and Gabrielle Capin for experimental help and data acquisition; Saara Forsström for comments on the manuscript; and Markus Innilä, Babette Hollmann, and Tuula Manninen for technical contribution and expertise. We thank the Biomedicum Imaging Unit and Genome Biology Unit for imaging equipment and expertise, Helsinki University Animal Center for animal husbandry and experimental support, and FIMM Technology Centre for untargeted metabolomics. We wish to acknowledge the following funding sources: Academy of Finland, Sigrid Juselius Foundation (A Suomalainen); University of Helsinki Doctoral Program In Biomedicine (O Ignatenko); Biomedicum Foundation (O Ignatenko); European Molecular Biology Organization (ALTF 1185-2017) (J Nikkanen); and Human Frontier Science Program Organization (LT000446/2018-L) (J Nikkanen).

### Author Contributions

O Ignatenko: conceptualization, resources, data curation, software, formal analysis, funding acquisition, validation, investigation, visualization, methodology, project administration, and writing—original draft, review, and editing.
J Nikkanen: data curation, software, formal analysis, investigation, methodology, and writing—original draft, review, and editing.
A Kononov: data curation, software, formal analysis, visualization, and writing—review and editing.
N Zamboni: data curation, formal analysis, validation, methodology, and writing—review and editing.
G Ince-Dunn: investigation and writing—original draft, review, and editing.

A Suomalainen: conceptualization, resources, data curation, formal analysis, supervision, funding acquisition, validation, investigation, visualization, methodology, project administration, and writing—original draft, review, and editing.

## Conflict of Interest Statement

The authors declare that they have no conflict of interest.

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
