## [Reviewer comments · Life Science Alliance]

Life Science Alliance

Mitochondrial spongiotic brain disease: astrocytic stress and harmful rapamycin and ketosis effect

Olesia Ignatenko, Joni Nikkanen, Alexander Kononov, Nicola Zamboni, Gulayse Ince-Dunn, and Anu Suomalainen

DOI: <https://doi.org/10.26508/lsa.202000797>

Corresponding author(s): Anu Suomalainen, University of Helsinki

Review Timeline:

Submission Date:	2020-05-27
Editorial Decision:	2020-06-16
Revision Received:	2020-06-23
Editorial Decision:	2020-06-26
Revision Received:	2020-07-03
Accepted:	2020-07-06

Scientific Editor: Reilly Lorenz

Transaction Report:

June 16, 2020

Re: Life Science Alliance manuscript #LSA-2020-00797-T

Prof. Anu Suomalainen-Wartiovaara
University of Helsinki
Research Programs Unit, Molecular Neurology
Biomedicum Helsinki
Haartmaninkatu 8
Helsinki FI-00290

Dear Dr. Suomalainen-Wartiovaara,

Thank you for submitting your manuscript entitled "Mitochondrial spongiotic brain disease: astrocytic stress and harmful rapamycin and ketosis effect" to Life Science Alliance. The manuscript was assessed by expert reviewers, whose comments are appended to this letter.

As you will see, the reviewers appreciate the interesting findings uncovered by your research, and they also note that your manuscript is really well written. While they raise some points to be considered, these involve potential edits to the manuscript text only and will not require additional experiments. We would therefore like to invite you to submit a revised version of the manuscript in response to these comments. When submitting the revision, please include a letter addressing the reviewers' comments point by point.

In our view revisions should typically be achievable in around 3 months, and in this case you may be completed with the text edits much sooner. However, we are aware that many laboratories cannot function fully during the current COVID-19/SARS-CoV-2 pandemic and therefore encourage you to take the time necessary to revise the manuscript to the extent requested above. We will extend our 'scooping protection policy' to the full revision period required. If you do see another paper with related content published elsewhere, nonetheless contact me immediately so that we can discuss the best way to proceed.

Please note that papers are generally considered through only one revision cycle, so strong support from the referees on the revised version is needed for acceptance.

Thank you for this interesting contribution to Life Science Alliance. We are looking forward to receiving your revised manuscript.

Sincerely,

Reilly Lorenz
Editorial Office Life Science Alliance
Meyerhofstr. 1
69117 Heidelberg, Germany
t +49 6221 8891 414
e contact@life-science-alliance.org
www.life-science-alliance.org

B. MANUSCRIPT ORGANIZATION AND FORMATTING:

Reviewer #1 (Comments to the Authors (Required)):

This is a very interesting and remarkable study that help us to better understand the cell and tissue-specific differences in mitochondrial diseases. Also, the study has a clinical importance since it shows that two promising therapies for mitochondrial diseases do not show benefits in the TwKOastro and TwKOneu mice, pointing out that the therapeutic responses in mitochondrial diseases are also heterogenous. The manuscript is very well written and the data support the conclusions.

Minor issues:

- An increase in the expression levels of the transsulfuration enzymes CTH and CBS is reported in the TwKOastro mice. Transsulfuration pathway provides the substrate to the mitochondrial hydrogen sulfide oxidation pathway. Could the stimulation of the transsulfuration pathway induce an adaptation on the mitochondrial hydrogen sulfide oxidation pathway? Could the authors comment this option?

- Discussion, p. 9, first paragraph: how can the data of the manuscript support that astrocytes boost glutathione biosynthesis? It seems that glutathione levels do not change in TwKOastro or TwKOneu mice, compared to control mice. Therefore, the results about glutathione metabolism differ to those published in the muscle/heart of the deleter mice (Nikkanen et al., CM 2016). Could the authors comment that difference?

- Discussion, p. 9, second paragraph: Kühl and colleagues (eLife 2017) reported an increase in proline biosynthesis rather than proline degradation. The levels of proline dehydrogenase were normal in the different mouse models tested in that study.

- Discussion, p. 10, first paragraph: It is interesting that both TwKOastro and Coq9R239X mice show astrogliosis and spongiosis and they do not show a positive response to rapamycin therapy. This could suggest that some common mechanisms may limit the rapamycin therapy in both mouse models. However, it is important to remark that TwKOastro mice do not show secondary CoQ deficiency according to the dataset (HMDB0002012). In the dataset, ubiquinone-1 must be ubiquinone-9.

Reviewer #2 (Comments to the Authors (Required)):

MS: LSA-2020-00797-T

The manuscript by Ignatenko et al. reports mitochondrial ISR signature in their CNS cell type specific model of mitochondrial DNA (mtDNA) depletion, and the response to treatment by rapamycin and ketogenic diet.

The models, neuronal specific KO and astrocyte specific KO of the mtDNA helicase Twinkle, have been described previously by the same group. Here, they report that the mitoISR response is cell type specific, because it is present in the brain of astrocyte, but not neuronal Twinkle KO. The astrocyte KO, however, does not show FGF21 or mTORC activation associated with mitoISR, unlike the muscle specific Twinkle KO. Both rapamycin and ketogenic diet exacerbated the disease phenotype in the two models.

It is intriguing that the mitoISR response is different in the two models, and the message that

treatment paradigms targeted for mitoISR amelioration, which worked in the muscle specific model, might actually be harmful for other cell types.

Overall, the manuscript is well written, and the results convincingly support their conclusions. Limitations of the use of cell specific knockout models are adequately discussed. There are, however, some aspects that deserve further discussion.

1. Based on their previously published findings, in the astrocytic Twinkle KO, significant depletion of mtDNA does not occur until late in the disease course (7-8 months), as opposed to the neuronal KO, which loses mtDNA earlier (3-4 months). However, the latter model does not display mitoISR. Since there is no correlation between mtDNA depletion and mitoISR, what is the driving factor for mitoISR? The authors ought to discuss the potential mechanisms. Is it the lack of energy, excess oxidative stress, or nucleotide imbalance?
2. In the neuronal Twinkle KO, there is no mitoISR signature. The authors should discuss the potential reasons why these two cell types respond so differently to the loss of Twinkle. Furthermore, they should comment on other potential metabolic and gene expression changes that underlie premature death of neurons in the neuronal KO model.
3. Is there evidence in these two models that energy metabolism shifts towards glycolytic dependence, as previously shown in other CNS cell type specific KO of mitochondrial OXPHOS proteins (i.e. COX10)?
4. In both models, there is neuronal death. The authors should elaborate on the mechanisms of cell autonomous and non-cell autonomous neuronal death, in relation to mitoISR activation.
5. Patients with MDS, have ubiquitous Twinkle mutations and multisystemic mitochondrial disease. Based on these and previous results, please comment on whether ketogenic diet or rapamycin are indicated under any circumstances for these patients.

Reviewer #3 (Comments to the Authors (Required)):

In this paper, the Suomalainen group shows another example of the amazing tissue specificity concerning the reaction of different cell types to mitochondrial dysfunction. Here, the mitochondrial DNA helicase Twinkle is ablated in neurons vs. astrocytes. Amazingly enough, it takes months until a phenotype develops, so studying the turnover of mitochondria in these two different cell types would be extremely interesting. However, this is not part of this current paper.

Since therapy is lacking for patients, they try to rescue the phenotype with two interventions which could be used in the clinic, namely a ketogenic diet or Rapamycin. The paper provides an extremely large and highly interesting set of data on the differential activation of the integrated stress response in those tissues and especially concentrates on measuring a large number of metabolites by latest state of the art techniques.

Although the paper does not provide any answer to the still amazing cell type heterogeneity, it provides an important source of information for future work on these two important brain cell types upon mitochondrial dysfunction.

Minor points:

I was trying hard to find minor points to correct but did not even find a typo.

Reviewer #1 (Comments to the Authors (Required)):

This is a very interesting and remarkable study that help us to better understand the cell and tissue-specific differences in mitochondrial diseases. Also, the study has a clinical importance since it shows that two promising therapies for mitochondrial diseases do not show benefits in the TwKOastro and TwKOneu mice, pointing out that the therapeutic responses in mitochondrial diseases are also heterogenous. The manuscript is very well written and the data support the conclusions.

Au: We would like to thank the Reviewer for the favorable and positive comments.

Minor issues:

- An increase in the expression levels of the transsulfuration enzymes CTH and CBS is reported in the TwKOastro mice. Transsulfuration pathway provides the substrate to the mitochondrial hydrogen sulfide oxidation pathway. Could the stimulation of the transsulfuration pathway induce an adaptation on the mitochondrial hydrogen sulfide oxidation pathway? Could the authors comment this option?

- Discussion, p. 9, first paragraph: how can the data of the manuscript support that astrocytes boost glutathione biosynthesis? It seems that glutathione levels do not change in TwKOastro or TwKOneu mice, compared to control mice. Therefore, the results about glutathione metabolism differ to those published in the muscle/heart of the deleter mice (Nikkanen et al., CM 2016). Could the authors comment that difference?

Au: We agree with the reviewer that the increased expression of enzymes of the transsulfuration and serine biosynthesis pathways might have pleiotropic effects on the cell metabolite fluxes. The responses may differ between the different tissues. However, steady-state metabolomics allows us to detect which pathways are affected in general. In Nikkanen et al., we found the serine flux to glutathione by metabolic flux analysis, by tracing isotope-labeled glucose carbons. In CNS, the complexity of cell types prevents currently fluxomics, and development of such methods are not in the scope of this paper. We have added this aspect in the discussion: page 9, paragraph 2.

- Discussion, p. 9, second paragraph: Kühl and colleagues (eLife 2017) reported an increase in proline biosynthesis rather than proline degradation. The levels of proline dehydrogenase were normal in the different mouse models tested in that study.

Au: We thank the reviewer for this note, and we have modified the sentence.

- Discussion, p. 10, first paragraph: It is interesting that both TwKOastro and Coq9R239X mice show astrogliosis and spongiosis and they do not show a positive response to rapamycin therapy. This could suggest that some common mechanisms may limit the rapamycin therapy in both mouse models. However, it is important to remark that TwKOastro mice do not show secondary CoQ deficiency according to the dataset (HMDB0002012). In the dataset, ubiquinone-1 must be ubiquinone-9.

Au: According to HMDB database, HMDB0002012 corresponds to ubiquinone-1, and therefore HMDB0002012 is annotated as ubiquinone-1 in our dataset, not ubiquinone-9. Interestingly, at the age of 3.2 months, the levels of ubiquinone-1 and ubiquinone-2 are decreased in TwKO^{astro} fed with a standard chow diet compared to Control mice (ubiquinone-1 $\log_2(FC) = -1$, $p_{\text{adjusted}} = 0.0000332$; ubiquinone-2 $\log_2(FC) = -0.99$, $p_{\text{adjusted}} = 0.0028$). We have now emphasized this point in discussion: page 10, paragraph 1.

Reviewer #2 (Comments to the Authors (Required)):

MS: LSA-2020-00797-T

The manuscript by Ignatenko et al. reports mitochondrial ISR signature in their CNS cell type specific model of mitochondrial DNA (mtDNA) depletion, and the response to treatment by rapamycin and ketogenic diet.

The models, neuronal specific KO and astrocyte specific KO of the mtDNA helicase Twinkle, have been described previously by the same group. Here, they report that the mitoISR response is cell type specific, because it is present in the brain of astrocyte, but not neuronal Twinkle KO. The astrocyte KO, however, does not show FGF21 or mTORC activation associated with mitoISR, unlike the muscle specific Twinkle KO. Both rapamycin and ketogenic diet exacerbated the disease phenotype in the two models.

It is intriguing that the mitoISR response is different in the two models, and the message that treatment paradigms targeted for mitoISR amelioration, which worked in the muscle specific model, might actually be harmful for other cell types.

Overall, the manuscript is well written, and the results convincingly support their conclusions. Limitations of the use of cell specific knockout models are adequately discussed.

Au: We would like to thank the Reviewer for the positive comments.

There are, however, some aspects that deserve further discussion.

1. Based on their previously published findings, in the astrocytic Twinkle KO, significant depletion of mtDNA does not occur until late in the disease course (7-8 months), as opposed to the neuronal KO, which loses mtDNA earlier (3-4 months). However, the latter model does not display mitoISR. Since there is no correlation between mtDNA depletion and mitoISR, what is the driving factor for mitoISR? The authors ought to discuss the potential mechanisms. Is it the lack of energy, excess oxidative stress, or nucleotide imbalance?

Au: We agree with the reviewer that knowing the signals of ISR^{mt} induction is an exciting, but yet an open question in the field. Mitochondrial DNA depletion induces changes in the levels of hundreds of metabolites, from amino acids to nucleotides and redox-signals and ATP, also affecting the expression of thousands of the genes. It is therefore challenging to confidently narrow down one or several signals driving the response in a CNS with a highly complex tissue type. We have now extended this point in discussion: page 9, paragraph 2.

2. In the neuronal Twinkle KO, there is no mitoISR signature. The authors should discuss the potential reasons why these two cell types respond so differently to the loss of Twinkle. Furthermore, they should comment on other potential metabolic and gene expression changes that underlie premature death of neurons in the neuronal KO model.

Au: We have now further elaborated on these points in the discussion: page 9, paragraph 2.

3. Is there evidence in these two models that energy metabolism shifts towards glycolytic dependence, as previously shown in other CNS cell type specific KO of mitochondrial OXPHOS proteins (i.e. COX10)?

Au: Induced glycolysis is a typical cell response to OXPHOS deficiency. We have not directly investigated whether the glycolysis is increased upon Twinkle knock out in our models, but previous studies suggest that astrocytes have higher basal level of glycolysis than neurons, and a higher capacity to upregulate glycolysis upon stress such as OXPHOS deficiency. (Supplie et al. J Neurosci 2017; Pacelli et al. Curr Biol 2015 ; Herrero-Mendez et al Nature Cell Biol, 2009). Indeed, if OXPHOS fails, the energetic crisis has to be compensated, and glycolysis is the key pathway to maintain ATP synthesis.

4. In both models, there is neuronal death. The authors should elaborate on the mechanisms of cell autonomous and non-cell autonomous neuronal death, in relation to mitolSR activation.

Au: We thank the reviewer for the interesting comment. We are unaware of published evidences linking the cell survival and ISR^{mt} induction - whether it is beneficial or deleterious is still undefined. This question was one of our motivations to study the effect of rapamycin, as in skeletal muscle, we have previously shown that inhibition of mTORC1 activity by rapamycin stopped disease progression and was even curative (Khan et al. Cell Metab 2017). However, as mTORC1 does not seem to similarly regulate ISR^{mt} in the brain, the regulatory loop remains open. Manipulating the ISR^{mt} components upon mitochondrial dysfunction to study the effects for cell fitness would potentially answer this question. This is a whole study on its own, and was judged not to be in the scope of the current study.

5. Patients with MDS, have ubiquitous Twinkle mutations and multisystemic mitochondrial disease. Based on these and previous results, please comment on whether ketogenic diet or rapamycin are indicated under any circumstances for these patients.

Au: Our study is, to our knowledge, the first one assessing the effects of rapamycin and ketogenic diet to the brain pathology caused by the mtDNA depletion, indicating deleterious results. In patients with devastating progressive MDS-related brain disease, such data are challenging to obtain, because differentiating the consequences of disease progression from treatment effect in autopsy samples is not possible. Our preclinical results presented here do not support rapamycin or KD for MDS patients, but indicate that the treatments may worsen the disease course. We have now modified this point in discussion to emphasize that caution is required when considering these treatment approaches in MDS: page 11, paragraph 1.

Reviewer #3 (Comments to the Authors (Required)):

In this paper, the Suomalainen group shows another example of the amazing tissue specificity concerning the reaction of different cell types to mitochondrial dysfunction. Here, the mitochondrial DNA helicase Twinkle is ablated in neurons vs. astrocytes. Amazingly enough, it takes months until a phenotype develops, so studying the turnover of mitochondria in these two different cell types would be extremely interesting. However, this is not part of this current paper.

Since therapy is lacking for patients, they try to rescue the phenotype with two interventions which could be used in the clinic, namely a ketogenic diet or Rapamycin. The paper provides an extremely large and highly interesting set of data on the differential activation of the integrated stress response in those tissues and especially concentrates on measuring a large number of metabolites by latest state of the art techniques.

Although the paper does not provide any answer to the still amazing cell type heterogeneity, it provides an important source of information for future work on these two important brain cell types upon mitochondrial dysfunction.

Minor points:

I was trying hard to find minor points to correct but did not even find a typo.

Au: We wholeheartedly thank the reviewer for such an enthusiastic reading of our manuscript.

June 26, 2020

RE: Life Science Alliance Manuscript #LSA-2020-00797-TR

Prof. Anu Suomalainen-Wartiovaara
University of Helsinki
Research Programs Unit, Molecular Neurology
Biomedicum Helsinki
Haartmaninkatu 8
Helsinki FI-00290
Finland

Dear Dr. Suomalainen-Wartiovaara,

Thank you for submitting your revised manuscript entitled "Mitochondrial spongiotic brain disease: astrocytic stress and harmful rapamycin and ketosis effect". We would be happy to publish your paper in Life Science Alliance pending final revisions necessary to meet our formatting guidelines.

- please provide source data for your Figure S2B (Figure EV2B currently)
- please consult our Manuscript Preparation Author Guidelines and order your manuscript sections accordingly
- please add a callout to Figure 1G
- please rename your EV figures as supplementary figures and adjust the callouts throughout your manuscript as Figure S1, s2, S3, etc.)
- please list 10 authors et al. in the references
- please provide your manuscript and tables in editable docx or excel format versions

A. FINAL FILES:

B. MANUSCRIPT ORGANIZATION AND FORMATTING:

Sincerely,

Reilly Lorenz
Editorial Office Life Science Alliance
Meyerhofstr. 1
69117 Heidelberg, Germany
t +49 6221 8891 414
e contact@life-science-alliance.org
www.life-science-alliance.org

July 6, 2020

RE: Life Science Alliance Manuscript #LSA-2020-00797-TRR

Prof. Anu Suomalainen-Wartiovaara
University of Helsinki
Research Programs Unit, Molecular Neurology
Biomedicum Helsinki
Haartmaninkatu 8
Helsinki FI-00290
Finland

Dear Dr. Suomalainen-Wartiovaara,

Thank you for submitting your Research Article entitled "Mitochondrial spongiotic brain disease: astrocytic stress and harmful rapamycin and ketosis effect". It is a pleasure to let you know that your manuscript is now accepted for publication in Life Science Alliance. Congratulations on this interesting work.

*****IMPORTANT:** If you will be unreachable at any time, please provide us with the email address of an alternate author. Failure to respond to routine queries may lead to unavoidable delays in publication.*******

DISTRIBUTION OF MATERIALS:

Again, congratulations on a very nice paper. I hope you found the review process to be constructive and are pleased with how the manuscript was handled editorially. We look forward to future exciting

submissions from your lab.

Sincerely,

Reilly Lorenz
Editorial Office Life Science Alliance
Meyrhofstr. 1
69117 Heidelberg, Germany
t +49 6221 8891 414
e contact@life-science-alliance.org
www.life-science-alliance.org